# Analysis of Meditation vs. Sensory Engaged Brain States Using Shannon Entropy and Pearson’s First Skewness Coefficient Extracted from EEG Data

**DOI:** 10.3390/s23031293

**Published:** 2023-01-23

**Authors:** Joshua J. J. Davis, Robert Kozma, Florian Schübeler

**Affiliations:** 1Department of Physics, Dodd-Walls Centre for Photonics and Quantum Technologies, University of Auckland, Auckland 1142, New Zealand; 2Department of Mathematics, University of Memphis, Memphis, TN 38152, USA; 3Kozmos Research Laboratories, Boston, MA 02215, USA; 4School of Informatics, Obuda University, H-1034 Budapest, Hungary; 5The Embassy of Peace, Whitianga, Coromandel 3591, New Zealand

**Keywords:** EEG, Shannon entropy, skewness, discrimination, cognition, emotion, spiritual values, meditation, awareness, intentionality

## Abstract

It has been proposed that meditative states show different brain dynamics than other more engaged states. It is known that when people sit with closed eyes instead of open eyes, they have different brain dynamics, which may be associated with a combination of deprived sensory input and more relaxed inner psychophysiological and cognitive states. Here, we study such states based on a previously established experimental methodology, with the aid of an electro-encephalography (EEG) array with 128 electrodes. We derived the Shannon Entropy (H) and Pearson’s 1st Skewness Coefficient (PSk) from the power spectrum for the modalities of meditation and video watching, including 20 participants, 11 meditators and 9 non-meditators. The discriminating performance of the indices H and PSk was evaluated using Student’s t-test. The results demonstrate a statistically significant difference between the mean H and PSk values during meditation and video watch modes. We show that the H index is useful to discriminate between *Meditator* and *Non-Meditator* participants during meditation over both the prefrontal and occipital areas, while the PSk index is useful to discriminate *Meditators* from *Non-Meditators* based on the prefrontal areas for both meditation and video modes. Moreover, we observe episodes of anti-correlation between the prefrontal and occipital areas during meditation, while there is no evidence for such anticorrelation periods during video watching. We outline directions of future studies incorporating further statistical indices for the characterization of brain states.

## 1. Introduction

It has been shown in the literature that prominent alpha-band activity occurs during meditative states and various spiritual practices associated with reports of inner peace [1,2,3]. Regarding meditation-induced alpha activity, Buzsáki remarked that “meditation is a widely practiced behavioral technique of tuning into your “inner self” but still remaining aware of the surrounding” [4]. It is frequently argued that we may feel a sense of calm when our brains are dominated by alpha oscillations [4]. Moreover, some studies have shown that people being measured via EEG with closed eyes show alpha oscillations, yet mostly in the occipital region [5]. To address the differences between meditative states and relaxed states with closed eyes, some authors have suggested that alpha or theta oscillations are also manifested in the prefrontal areas, in addition to the alpha oscillations observed in the occipital region during closed eyes [5,6,7]. Further, other bands may be involved in meditative states, as described by [8] in studies with experienced Buddhist meditators, who show an increased amplitude of gamma oscillations in tasks involving the practice of compassion. It is important to mention that previous work on Zen meditation techniques with practitioners of varying skill levels and years of experience show predominantly alpha oscillations. Moreover, after some meditation time (~27–28 min), theta oscillations appear for several participants, which in occasions may be associated with drowsiness. Even though these early studies [9,10] are precursors to modern studies [11] using the art of encephalography [12], they still provide relevant information on meditative states.

The classification of different biological signals, in particular brain signals, has been attempted via various quantitative techniques and methods [13,14,15,16,17,18,19], and there is no consensus on the most effective methods of classification. One of the limitations in using EEG measurements is spatial resolution, which can be compensated for by the use a high-density array of electrodes of the kind that we employ [13,18,20,21]. This is an important technical and technological consideration that may facilitate the proper classification of different cognitive states associated with different areas of the brain.

Another unresolved issue is related to the observed variability in brain dynamics associated with various meditation techniques and different levels of mastery of such techniques, together with the fact that meditative techniques are usually performed with eyes closed, known to produce alpha oscillations in the occipital area, which increase the difficulty of reliably classifying brain dynamics for different meditation techniques and discriminating them from a closed-eyes modality only [6,7,17,22].

In previous studies, we have analyzed the differences in brain dynamics between open-eyes and closed-eyes modalities using various electrode arrays, including 256 electrodes covering the scalp and a 48-electrode Mindo head band placed across the forehead of the participants [11,20,21]. These studies serve as a baseline reference for the work presented in this report, and they demonstrate remarkable differences between closed-eyes and open-eyes conditions. However, the difference between meditation and open-eyes experiments were relatively small when using Mindo, and the studies remained inconclusive regarding the distinctions between meditative brain states and states with closed eyes. Other studies show such difference [23,24], thus an important objective of this work is to investigate these effects deeper.

Finally, it is important to mention that when performing audiovisual activities, emotions may be elicited [25] that will be reflected in brain dynamics [26], and this may be difficult to discriminate and separate (via EEG signals only) from the visual and auditory effects produced by environmental stimuli [27]. This is relevant in our study since we measured brain dynamics in multiple modalities.

In this study, we invoke the methodologies described in [5,11,20,28], by applying statistical evaluation of new experimental EEG data, obtained in Ian J. Kirk’s Lab, Centre for Brain Research at the University of Auckland in New Zealand [29,30,31], during a period of about 3 months of data acquisition in 2 modalities and with 20 participants, as part of a larger study with 6 modalities. The modalities analyzed and studied here are Meditation-Relaxation (MED) versus Video Watching (VDO) for 11 Meditators and 9 Non-Meditators.

We use the Shannon entropy index (H) and Pearson’s 1st skewness coefficient (PSk) to characterize different brain states. Employing H and PSk are part of a broader methodology that we have developed previously [5], and the present study allows testing the classification power of these indices with a larger database of participants. In this work, we focus on these two indices, but our larger approach includes other measures, as well, such as power, dominant frequency [11], analytic amplitude, analytic phase, instantaneous frequency and pragmatic information indices [12,20,21]. The Shannon entropy index (H) has become popular in EEG studies [32,33,34], including various variants, e.g., sample entropy, fuzz entropy and differential entropy, which differ among studies, with different meanings in some cases. Other studies have shown the use of higher order statistics, such as the PSk, in order to describe the non-Gaussian feature of EEG [35].

We observed a clear difference among the statistical features extracted for the two groups of participants and between the modalities. We investigated if meditative states (MED) are less entropic and more ordered than busy states requiring more information processing and energy consumption, as, for example, in the modality of VDO. In general, we expect that meditative or relaxing states ought to differ significantly from more busy states, such as watching a pleasant musical video with changing ambiguous images that engage with the visual and auditory brain systems. We propose that the power spectrum associated with brain dynamics measured in the VDO modality may manifest a more diverse Power Spectrum, with frequencies in the Theta, Beta and Gamma bands, and less prominence of the Alpha band in different brain areas [36,37], significantly contrasting with meditative or relaxed states, where the Alpha band becomes dominant for certain brain regions.

We performed an extended analysis of the EEG array data in the MED and VDO modalities, showing that both the H and PSk indices are able to characterize brain dynamics in different modalities, associated with different brain, psychophysiological and inner states. Statistical tests have been performed to successfully discriminate between the MED and VDO modalities, Meditator and Non-Meditator participants, and dynamics of the prefrontal and occipital areas, based on H and PSk indices. We intend to explore in the future the use of a broad range of indices, while here we focus on the behavior of H and PSk statistical features.

## 2. Materials

The EEG technology used for this study was a HydroCel Geodesic Sensor Net (HCGSN), 128 electrodes dense-array electroencephalography (EEG) produced by Electrical Geodesics, Inc., USA [38]. Data were recorded via Net Station 4.4.2 at a sample rate of 1000 Hz. Due to the highly efficient and accurate “sigma-delta” type A/D converters for each channel, there is no need for multiplexing on this platform. The results obtained from the 128 electrodes were recorded in a 128-vector that, after preprocessing, was reshaped into a 12 × 12 matrix (A) representing the different brain areas. Some electrodes were doubled in order to fill the matrix. Positions A_1,3_ and A_1,10_ are empty as reference points for the position of the pre-frontal cortex; see Figure 1.

We have expanded on earlier studies [5,11] to include twenty (20) participants that were measured in six (6) modalities: meditation (MED), scrambled words (WORDS), ambiguous images (IMG), mental arithmetic (MM), positive affirmations (SENT) and video watching (VDO). The order in which we measured the modalities was selected to allow participants to prepare in meditative or relaxed states first, to be able to address the more demanding tasks with better cognitive skills. By the time they reached the modality of VDO, around 30 min after the modality of MED, they were certainly in a state quite different than meditation. If the effects of meditation would propagate and influence the brain dynamics of the participants when in modality VDO, we would have the opportunity to observe that when analyzing and comparing the results between meditators and non-meditators. We used E-Prime 2.0 to program the experimental setting and the presentation of stimuli, as well as relevant events, such as ‘start’ and ‘press key’.

Here, we only describe the 2 modalities relevant to this study, as follows:**Meditation (MED):** Participants were asked to engage in the meditation of their choice for 7 min with their eyes closed. Alternatively, if people were unfamiliar with meditating, they were asked to relax with their eyes closed. After the proper preparation, each participant pressed the space bar key on the keyboard to signal the beginning of the meditation period, which continued until the preprogramed end signal informed them of the end of the session.**Video (VDO):** In this final modality, the participants were presented with a video containing a sequence of ambiguous images with the song ‘Imagine’ by John Lennon, playing throughout the video. The ambiguous images aimed to evoke mental responses similar to the well-known Necker cube. The ambiguous images used in this task were designed by Oleg Shupliak, and they can be found in [39]. The duration of this experiment was 1 min and 50 s for each participant. There was no task the participants were asked to perform besides watching the video and listening to the song. After reading the instructions, the participants pressed the space bar key on the keyboard to signal the beginning of the video-watching period, until the video finished playing.

For the experiments, the participant sat inside a Faraday chamber, positioned in a comfortable chair, facing a computer screen with a standard computer keyboard in front of them. Participants were instructed to use only the space bar and the numerical keypad, and to restrict head and eye movement as much as possible to minimize artifacts in the EEG data recording. An impedance check was done prior to the beginning of the experiment.

Among the 20 participants (P1 to P20), all healthy, there were 11 males and 9 females, varying in age from 23 to 64 years for non-meditators and from 38 to 63 years for meditators. We relied on the information that the participants provided in interviews and conversations with them, and on their personal details and informed consent documentation. Some were regular practitioners of meditation, while others had no or little previous experience in meditation. We grouped participants in the *Meditator* group if the participant had been meditating for at least 2 years on a regular basis, at least 5 days a week. Eleven of the participants fell into this group, while the remaining nine participants were placed in the *Non-Meditator* group.

## 3. Methods

### 3.1. Preprocessing

For the EEG signal processing, we followed the methodology introduced in [11]. A notch filter for 50 Hz was applied, together with a finely tuned filter and detrend algorithms as a means to remove artefacts derived from blinking and other body movements, resulting in a reliable and ‘clean’ data set with a broad spectrum between 2–48 Hz.

We computed the temporal power spectrum (PSD_t_) over a time window of 500 ms for each electrode in the 12 × 12 matrix, where the index ‘t’ in PSD_t_ refers to time. After that, we evaluated two (2) information measures, namely, the Shannon entropy or diversity index (H) and Pearson’s first skewness coefficient (PSk), both based on power and frequency band, taken as a histogram and empirical probability function. Both H and PSk have shown to be good indices to classify the different brain dynamics [5].

In order to match the recorded event times, such as ‘press key’, for example, with the recorded EEG data, two (2) timing corrections were applied as follows:8 ms for anti-alias filter, which essentially accounts for the required time of the amplifier to do the conversion;14 ms for the screen refresh rate, adding to a total of a 22 ms shift to match the recorded event markers with the actual event time.

In general, all adjustments and preprocessing apply equally to all channels to assure that all signals remain the same length.

### 3.2. Entropy and Information Theoretical Indices

As reported in previous studies [5], we have aimed at devising information and entropy measures that would allow for the investigation of meaning structures as manifested in brain dynamics. Meaning structures have been associated to: (a) Phase Transitions from disorder to order and vice versa, in nonlinear dynamics, and (b) Superstructures, as shown by [40] and [41], respectively. In addition, phase transitions in cortical brain dynamics have been described as the emergence and disintegration of large-scale functional brain structures at criticality [42,43], which are the fundamental dynamics needed to understand the Cycle of Creation of Knowledge and Meaning in the context of intentional actions and values-based decision making, as treated by [44], this being the main interest of our larger study and work that include all of the above mentioned six (6) modalities.

Here, we expand on 2 types of information and entropy measures, H and PSk, respectively, which are derived from the normalized Power Spectrum when treated as an empirical probability distribution, as depicted in Figure 2. The PSk coefficient is viewed as an information measure, and it is derived from the mean, standard deviation and mode (dominant frequency) of the normalized Power Spectrum. We use six frequency bands, as shown in the table in Figure 2, in order to derive the probability distribution for a typical experimental setup to illustrate the methodology. In this illustration, we display some possible values that allow performing the computations meaningfully for this example; the full details are provided in the next section.

In our estimation, both the indices H and PSk, together with other indices, such as the pragmatic information index [45], may allow for the detection of meaning and knowledge creation as manifested in brain dynamics, and are needed to advance our understanding of meaning, knowledge and intentionality. For this study, we focus on:3.Entropy measure (H), as introduced by Shannon [46], providing us with the degree of randomness in an EEG signal.4.PSk as a measure of information derived from the frequency distribution structure represented by the degree of asymmetry, which was derived by [47] and discussed by [48]. The version of the PSk (1st order skewness coefficient) described and formulated in [49] in terms of the mean, standard deviation and mode (dominant frequency or frequency band) is used for this study.

The computations for H and PSk are treated in the following section.

### 3.3. Computation of the H and PSk Indices

Here, we present the equations to compute *H* and *PSk*, following [5]. For *H* we have:(1)$$H=-\overset{n}{\underset{i=1}{\sum }}p_{i}*\;log_{2}\left(p_{i}\right)$$
where $p_{i}=\frac{PW_{i}}{TP}$, *PW_i_* corresponds to the power of frequency band ‘*i*’, and *TP* is the total power computed as:(2)$$TP=\overset{n}{\underset{i=1}{\sum }}PW_{i}$$

The equation for the computation of *PSk* is:(3)$$PSk=\left|\frac{\left(Mean_{PSDt}-Mode_{PSDt}\right)}{SD_{PSDt}}\right|$$
where *PSD_*t*_*, as mentioned above, is computed as:$$PSDt≐\;PW_{i}\;\left(FB_{i}\right),\;\forall \;i$$
where *SD_PSD_*t*__* represents the standard deviation of the *PSD_*t*_*, and where power is taken as a function of frequency and, therefore, frequency band, described here as *PW_i_*(*FB_i_*).

When computing H, we used the probabilities p_i_ derived from the *PSD_*t*_*, where the number of a particular band is described by a fixed number ‘*i*’ for all *PSD_*t*_* for any participants and modalities, as follows:$$PSDt≐\;PW_{i}\;\left(FB_{i}\right)\;\forall \;i=4,\;5,\;6,\;\ldots ,\;48\;Hz$$

As described in previous work, we may need to adjust H to a new value H_c_, as described by [50], something left outside of the scope of our study. However, H_c_ may be useful when comparing brain dynamics for different brain areas in different bands, participants and modalities that depend on unique and specific probability distributions. This analysis included every electrode, as well as brain areas represented by their corresponding set of electrodes, from where we derived H and PSk indices for our analysis, as shown in Figure 3.

### 3.4. Analysis of Multi-Variate EEG Data

Here, we present the method used to perform the statistical analysis of the EEG array data. We provide intervals of confidence per group and the modality for H and PSk, based on the mean values per electrode, per participant, per time window ‘t’ of the Power Spectrum. We show the general definitions and formulas, as follows:

$H_{t,e\;}^{p,m}$ is defined as a value of *H* derived from the *PSD* in window *t*, for participant *p*, in modality *m*, for electrode *e*, where the number of windows (*NW*), in which we compute *PSD*_t_, is derived from the equation $NW=\frac{L}{500}$, where *L* is the time length (in ms) of a particular experiment for participant *p* in modality *m*, and where the length for each window *t* equals 500 ms.

We define the mean value of *H* over all windows per electrode, per participant, per modality, as follows:$${\bar{H}}_{e}^{p,m}=\overset{NW}{\underset{t=1}{\sum }}\frac{H_{t,e\;}^{p,m}}{NW}$$

Next, we define the mean value ${\bar{H}}_{e}^{p,m}$ over all electrodes, per participant, per modality:$${\overset{=}{H}}^{p,m}=\overset{128}{\underset{e=1}{\sum }}\frac{{\bar{H}}_{e}^{p,m}}{128}$$

Finally, we define the mean value of *H* per group, per modality, as follows:$${\bar{H}}_{g}{}^{m}=\sum_{p=1}^{npg}\frac{{\overset{=}{H}}^{p,m}}{npg}\;\forall \;g=1,\;2$$

In the above formula, *npg* equals the number of participants per group, 11 for the *Meditator* group (*g* = 1) and 9 for the *Non-Meditator* group (*g* = 2). Similarly, we computed the standard deviation for the value of H per group, per modality, and we have labeled it as $Hs_{g}{}^{m}$, from which we can derive the interval of confidence for ${\bar{H}}_{g}{}^{m}$, as follows:$${\bar{H}}_{g}{}^{m}\;\pm \;t_{\propto =0.05,\;npg}*\frac{Hs_{g}{}^{m}}{\sqrt{npg}}$$

It is important to note that the above formulas similarly apply to the computation of ${\bar{PSk}}_{e}^{p,m},\;{\overset{=}{PSk}}^{p,m},\;{\bar{PSk}}_{g}{}^{m}\;and\;PSks_{g}{}^{m}.$

## 4. Results

### 4.1. Qualitative Analysis of the EEG Data

We introduce the results of the analysis of the recorded brain dynamics data for the 2 groups of participants for the modalities MED and VDO, for the array of 128 electrodes. We present several figures that qualitatively and quantitatively illustrate our findings, by showing the mean values and landscapes associated with H and PSk for each participant, for the two modalities. These landscapes should be read as shown in Figure 3 and will apply to all landscape plots in various subsequent figures.

In Figure 4, we can observe that the mean values for H and PSk in the modality MED are significantly lower when compared to the mean values in modality VDO for most of the *Meditator* group. Within the modality of MED, participants P8 and P17 show relatively higher mean values for both indices. Within the modality VDO, participants P17 and P20 display similar mean values as in the MED modality; however, participant P20 shows significantly lower mean values than P17.

At this stage, we observe an average tendency in PSk and H values, indicating that brain dynamics in the modality MED are less entropic or more ordered (structured) than brain dynamics in the modality of VDO for the *Meditator* group. The question is raised, could it be possible that meditators may carry their relaxed state into other activities?

Next, we perform a similar analysis for the *Non-Meditator* participants, revealing certain similarities and differences between the two groups.

In Figure 5, we can observe that the mean values for H and PSk in the modality MED for the *Non-Meditator* group are significantly lower when compared to the mean values in the modality VDO for most *Non-Meditator* group members. Within the modality MED, participant P14 shows relatively higher mean values than the rest of the *Non-Meditator* group members for both indices. However, within the modality VDO, participant P14 displays still high values as in the MED modality, yet similar mean values to the rest of the *Non-Meditator* group members in the VDO modality.

We observe a tendency for participants who display relatively low values for H and PSk in the modality MED to also display relatively lower mean values when compared to the other participants for both the *Meditator* and *Non-Meditator* groups. This suggests that meditators may carry the effect of deep relaxation and meditation into other activities, at least for a period, until the effects of meditation are lost in social dynamics.

Generally speaking, when we compared the values for the *Meditator* vs the *Non-Meditator* groups in both MED and VDO, the *Meditator* group members display lower mean values for H and PSk, to be described in the next subsection.

There are, however, in the modality MED, *Non-Meditator* members (P11 and P18) who, based on their H and PSk mean values, could fall in the *Meditator* group and vice versa (P8 and P17). Such exceptions can be observed in the VDO modality.

It is interesting to note that most participants show similar patterns for both H and PSk. For example, if we look at the pattern for P7 in the modality MED, we can clearly observe lower mean values for both H and PSk in the occipital region and similar higher ones in the other regions. It is important to note that the overall variability of H and PSk mean values is greater for both groups in the VDO modality than in MED, most clearly observable for the *Non-Meditator* group.

Further, we observe an overall tendency of lower H and PSk values for many of the participants in the central frontal and central posterior regions, particularly in MED. One possible explanation for this may be the lack of visual stimuli (closed eyes), which is known to cause activity around the posterior part of the cortex, as shown in previous studies, where participants in an eyes-closed modality show alpha dominance in the posterior parietal and occipital cortex, as measured via EEG [51,52,53]. This might be a reason why participants in both the *Meditator* and *Non-Meditator* groups show alpha dominance in the modality of MED, since this was done with closed eyes. Such alpha dominance is associated with low values of H and PSk, as treated by [5].

Overall, our results suggest that alpha dominance, from where we derive low mean values for H and PSk in the MED modality, is observed mainly in the occipital cortex for both the *Meditator* and *Non-Meditator* groups, and similarly in the pre-frontal and frontal areas of the *Meditator* group members.

This difference may prove useful when characterizing the different brain dynamics associated to these two groups, when we take the behavior of different brain regions together. Aside from H and PSk, other indices and information measures derived from the PSD_t_, such as the Dominant Frequency band, for example, need further investigation and analysis, which is left for future studies.

Finally, to conclude this first qualitative analysis, we must note that, generally speaking, the H and PSk mean values for the large majority of participants are significantly smaller in the modality MED when compared to the VDO modality. In general, lower H and PSk mean values are observed in the modality VDO for participants in the *Meditator* group, when contrasted with participants of the *Non-Meditator* group.

Next, we support our qualitative analysis and observations with the appropriate quantitative statistical measurements for H and PSk, obtained for both the *Meditator* and *Non-Meditator* groups in both modalities, MED and VDO.

### 4.2. Detailed Quantitative Analysis of Brain Dynamics

It is important to note that, throughout the paper, we have used, in a general manner, the labels H and PSk to denote the values of ${\overset{=}{H}}^{p,m}$, as described in Section 3.4. This has been the use when computing the landscapes in Figure 4 and Figure 5 previously, and this will be used for the values of H and PSk later in this Section. However, H and PSk refer to ${\bar{H}}_{g}{}^{m}$ when computing the intervals of confidence displayed in Table 1 and 2 and Figure 6 and Figure 7.

As we have said before, when comparing the overall H and PSk mean values for each modality, the modality MED displays the smallest values for both indices, while the modality VDO displays the largest H and PSk mean values for both *Meditator* and *Non-Meditator* groups, as can be clearly observed in Table 1 and Table 2 and Figure 6 and Figure 7.

When we take a closer look at the mean values of H in the MED modality, there is a difference that may account for the distinct brain dynamics between the Meditator and Non-Meditator groups. This difference is undetected by the PSk index, as shown in Figure 6, due to a very small difference in mean values, buried in a significant amount of variability.

This is why we must use both indices combined when aiming at classifying different brain states for different participants in different modalities. As we have proposed previously, we will need an additional set of indices [11,45,54], such as *Total Power*, *Dominant Frequency Band*, *Planckian Information* and *Pragmatic Information*, to make our methodology very robust, something we intend to advance in new studies.

When combining the two indices, it becomes clear that there is a distinct difference between Meditators and Non-Meditators in brain dynamics, and that H and PSk taken together are a useful element to analyze and differentiate between various cognitive states in different modalities for different groups of participants, as shown in Figure 7.

If we remember the qualitative analysis, as shown in the different landscapes in Figure 4 and Figure 5, it becomes very relevant in making distinctions for the brain dynamics occurring in different areas of the brain in the different modalities, for the different participants and groups of participants.

Student’s *t*-test of the zero hypothesis (H0: μ1 = μ2) was performed for the MED and VDO modalities, having 837 and 227 data points, respectively. The tests demonstrated a significant difference between the mean H values between modalities MED and VDO, with *p*-values below 0.05, both for Meditators and Non-Meditators, indicating that the brain dynamics associated to these modalities are significantly different. The same observation is valid for the PSk index, with *p* = 0.03. Thus, H and PSk can serve as valuable indices to discriminate between the MED and VDO modalities.

Next, the discrimination between Meditators and Non-Meditators is studied. Table 3 shows the results of the t-test with unequal variances regarding the zero hypothesis (H0: μ1 = μ2), comparing the group of Meditators (1) vs Non-Meditators (2), for each modality (MED and VDO) and indices H and PSk.

When we compare Meditators and Non-Meditators for the modality VDO, we are led to accept the null hypothesis H0 based on both H and PSk. Thus, the VDO modality does not allow discrimination between Meditators and Non-Meditators.

However, for the modality MED, the null hypothesis H0 is rejected based on H, while it is accepted based on PSk. This indicates that the H index allows discriminating between the brain dynamics of Meditators versus Non-Meditators in the MED modality. Such an outcome may support our expectation that the brain state corresponding to “relaxation only” is different from the meditation state.

This raises the question, which index should we trust the most: H, PSk or both? When we consider both together, which has been our approach, we need to explore more deeply what happens in the prefrontal area of the brain in contrast with the occipital area. The brain dynamics of the prefrontal region of meditators could be a decisive factor for differentiation, while the occipital region may behave similarly for both groups when eyes are closed, shown as steady and dominant Alpha oscillations. These types of brain dynamics have been shown to be associated with lower values for H and PSk [5]. To explore these issues, we analyze the H and PSk values in the prefrontal and occipital brain dynamics in Section 5 via another set of hypothesis tests.

Still, in this section, we carried out further visual inspection of the behavior of H and PSk for the different electrodes and their associated brain regions, as shown in Figure 8. This was done as an example for two (2) participants and will certainly require a robust quantitative analysis per brain region for all participants, which remains outside of the scope of this paper.

In Figure 8, we display scatter plots between H (X axis) and PSk (Y axis), with their respective histograms on each axis. The empirical distributions represented by the histograms are different for each participant in each modality, and they may, when properly statistically assessed, differ in skewness and kurtosis, depending on the brain dynamics of participants in the different modalities and groups.

When we look at the scatter plot distribution, we may start to identify apparently nonrandom structures or chaotic attractors, which display different patterns for different participants in the different modalities. In the case of participant P3 in the modality MED (Figure 8), we observe a bimodal structure, which may be associated with different behaviors in different areas of the brain.

In order to investigate the different structures associated with different areas of the brain, we produced Figure 8, where we show the frontal, middle (center) and posterior (back) areas of the brain as clusters of different colors—red, blue and black, respectively.

Additionally, in Figure 8, we can identify in the axis of the PSk index how distinct the frontal region behaves when compared to the posterior region, as quantified by the PSk mean values per electrode, as well as the mean values per region, shown as large circles. The differences may appear small upon visual inspection, since we have to preserve all graphs in the same scale range (H: 3 to 5.5 and PSk: 0 to 1) to make it possible to compare the range of values across regions, modalities and participants. Nonetheless, the differences are noticeable, and this issue will be the objective of future studies.

As in the above analysis, we again observe the differences in values and structures for modality MED when compared to modality VDO.

We have illustrated that different participants show distinctly different configurations of structure when plotting each electrode for H against PSk. When we looked at all participants, groups and modality plots, we found most of these structures to be either divided into two distinct clusters or displaying a long stretch with a distinct tendency for higher PSk mean values as H mean values increase, as can be observed for P3 in the modality VDO in Figure 8, for example.

When looking at the plots for each and every participant (similar to the plots in Figure 8), we recognize that while for some participants, the mean values cluster together (e.g., P4 in VDO in Figure 8), for many participants, the different regions are clearly separated, as, for example, in P3 and P4 in the MED modality, as shown in the same figure.

It is interesting to note that, for many participants, the blue- and black-colored values associated with the central and posterior areas of the brain, respectively, tend to cluster together, while the red values associated with the frontal area appear to be distinctly separated, as, for example, for P3 in the modality MED. Further analysis of the structures and distributions will be required to reach some robust conclusions.

Overall, we can state that brain dynamics are unique for each participant, and when represented via the H and PSk values, certain patterns and tendencies can be observed for the different groups in the different modalities. Participants of the *Meditator* group tend to display smaller H and PSk values for both modalities than members of the *Non-Meditator* group, and participants of both groups display smaller values for the modality MED than for the modality VDO.

## 5. Discussion

### 5.1. Discrimination across Modalities and Brain Regions for a Representative Participant (P10)

To provide a deeper understanding of the spatio-temporal EEG dynamics, we analyze H and PSk as stochastic processes evolving in time, each 0.5 s. In the analysis, we use the correlation coefficient between H and PSk in the different modalities (MED, VDO) for different brain areas and different participants. Further, in order to describe internal states that may be derived from meditation and other spiritual practices, such as inner peace, we will require the application of a combination of 3rd-, 2nd- and 1st-person perspective science.

Figure 9 depicts the temporal evolution of H and PSk during the MED and VDO modalities, for participant P10. In MED modality, the OCC region shows lower mean values than the PF region, for both H and PSk. However, for VDO, the PF and OCC areas show similar mean values. There are higher values for both H and PSk in the VDO modality as compared to the MED modality, in both the PF and the OCC regions. The differences between modalities are more prominent in the OCC region, which can be explained due to the closed eyes in the MED modality.

Moreover, for MED, we observe that the time series for both H and PSk appear to be inversely correlated, showing some relatively long-term cyclical oscillations, something that could be related to “warming up” and “cooling down” periods in meditative states, as well as internal and environmental emotional and cognitive signals. There is no evidence for such anticorrelation periods in the VDO modality. The vertical arrows with stars in Figure 9, subplots (a) and (b), indicate anticorrelation events between the occipital and prefrontal areas during MED. This observation needs future investigation.

In Figure 10, we plot the histograms of H and PSk for both areas of the brain for P10, where we can observe, for example, that for the modality of MED, the values of H show a bias towards larger mean values for the PF area than for the OCC area, while in the modality of VDO, this bias is absent, and both areas show similar distributions. We foresee that, in future studies, more analysis of this kind may contribute to our learning process about brain dynamics.

In Figure 11, we show the coefficient of correlation (r) between H and PSk in the different modalities, for both the PF and OCC areas of the brain, for P10. We observe significantly different behaviors in the scatter plots, with their respective mean r values and confidence intervals at 95% significance [55].

We note that r can be used as another, perhaps more robust classifier in future studies, as we can observe that r shows statistically significantly different values for the different brain areas and modalities studied; see Table 4. This will require further investigation.

Correlations, both negative and positive, can be observed between electrodes, as displayed in the correlation matrices for H and PSk in MED and VDO, including the 128 electrodes, as shown in Figure 12. Generally speaking, electrodes in nearby areas of the brain show higher values than electrodes far apart, something we expected. However, for this participant, P10, when we look at the display of H in MED in Figure 12a, we can observe negative correlations between the OCC and the PF areas. As for the VDO modality, in Figure 12c, no negative correlation is observed between the OCC and PF areas.

All of this suggests the negligible or non-influence of meditative states when watching a video around half an hour after meditation. However, trained meditators could actually carry with them a tendency towards Alpha dominance characterized by lower H and PSk values, requiring additional studies of the prefrontal and occipital regions. Future studies on these issues may add value to the understanding of brain dynamics for different participants in different modalities for different brain regions.

Taken together, the results obtained for a single participant (P10) indicate significant differences between the brain dynamics observed in the MED and VDO modalities, as measured by H and PSk. To elaborate on these initial findings, and to make them conclusive, we need a larger database of participants.

### 5.2. Discrimination Results in Populations of Meditators and Non-Meditator Participants

In order to further assess the potential of the H and PSk indices discriminating between Meditators and Non-Meditators in both the MED and VDO modalities, we tested for the difference between brain dynamics in the PF and OCC areas using indices derived for all 20 participants. In Table 5 and Table 6, we present the results of the tests, with the zero hypothesis that the means (μ1 and μ2) of the two population distributions are the same. We compared the modalities MED and VDO and the groups of Meditators and Non-Meditators, for the PF and OCC areas, using indices H and PSk.

In Table 5, using the H index, the *p*-values are below 0.03 in the MED modality, which indicates that the null hypothesis H0 is to be rejected for both areas (PF and OCC). Rejection of the null hypothesis for the MED modality indicates that the Meditator and Non-Meditator group show different brain dynamics, as represented by H, for both the PF and the OCC areas. In the VDO modality, this behavior is reversed, and H0 is accepted for both areas. This shows that the brain dynamics of the Meditator and the Non-Meditator groups can be discriminated from each other in the MED mode using the H index; however, they are not distinguishable in the VDO modality.

In Table 6, results with the PSk index are shown. Based on data obtained over the PF, the zero hypothesis is rejected for both the MED and VDO modes, with *p* values below 0.05. However, the zero hypothesis is accepted, based on the OCC area data for both modes. This means that PSk is able to distinguish Meditators from Non-Meditators using PF data, but it is not discriminative using OCC data.

These tests indicate that the prefrontal area behaves differently for Meditators than for Non-Meditators in the Modality MED, based on both the H and PSk indices. This seems to suggest that we can rule out the possibility that the values observed in the MED modality for Meditators are due to closed eyes only. In the same modality of MED, the test based on H is rejected, while it is accepted for the OCC area when based on PSk.

This leaves us with the uncertainty of whether meditative practices may impinge on different areas of the brain, such as the occipital area, enhancing alpha dominance leading to low H values, with long-lasting effects. This issue remains inconclusive and requires future studies.

Similarly, in the VDO modality, when comparing both groups, the test is rejected for the PF area based on PSk, while accepted based on H; hence, it remains inconclusive if watching a video has the same effect on Meditators and Non-Meditators in the PF area. It seems to be the case that watching a video shows the same behavior, based on the statistics of both H and PSk for the OCC area, since the null hypothesis H0 is accepted for both indices.

## 6. Conclusions

Extending on previous studies [11,56], we have presented the complementary use of the Shannon Entropy Index (H) and the Pearson’s Skewness Coefficient (PSk) to classify brain dynamics for different brain regions, different participants and different modalities. In the future, we intend to incorporate more indices for the classification and discrimination between brain states, to make our methodology more robust, and contrast it with other classification algorithms already tested and available in the literature. Based on our observations and results, we have the following conjectures:We conjecture that more *relaxed states* showing alpha dominance, accompanied with lower values of H and PSk, are achieved by: (1) masterful meditators, (2) people who practice relaxation techniques and (3) people who are naturally more relaxed (less stressed), who might be able to mitigate environmental signals and demands when existing in such *relaxed* emotional and coherent mental states. This we can derive from the data associated with the modality VDO when contrasted with the one of MED for both groups: *Meditator* and *Non-Meditator*. It is relevant to note that the *Meditator* group showed lower values for H than the *Non-Meditator* group, which indicates that meditative states are more likely different than relaxed states when participants have their eyes closed.We conjecture that meditators may carry these relaxed states into other activities, possibly due to the lasting psychophysiological effects derived from meditative practices [28,57,58,59] translating and continuing into other areas of life. This will require further investigation and studies with a larger sample size.When comparing the overall mean values for each modality, MED displays the smallest values of H and PSk, and VDO displays the largest for both groups. The overall distribution and values, however, are significantly different. These findings indicate that there is a distinct difference between meditators and non-meditators in brain dynamics, and that H and PSk taken together are a useful element to analyze and differentiate various cognitive states. Statistical hypothesis tests indicate that the H index is useful to discriminate between *Meditator* and *Non-Meditator* participants during MED over both the PF and OCC areas (*p* = 0.03), while the PSk index is useful to discriminate *Meditators* from *Non-Meditators* based on the PF areas for both MED and VDO (*p* = 0.05).

Finally, remembering the big picture that motivates our research, we foresee that a deeper understanding of the relationship between brain dynamics, cognition and emotions will add value to the understanding of meaning and values, both behavioral and spiritual, when performing intentional actions and making decisions [44,60,61], and our methodology, together with others, helps researchers to progress towards these goals.

## Figures and Tables

**Figure 1 sensors-23-01293-f001:**
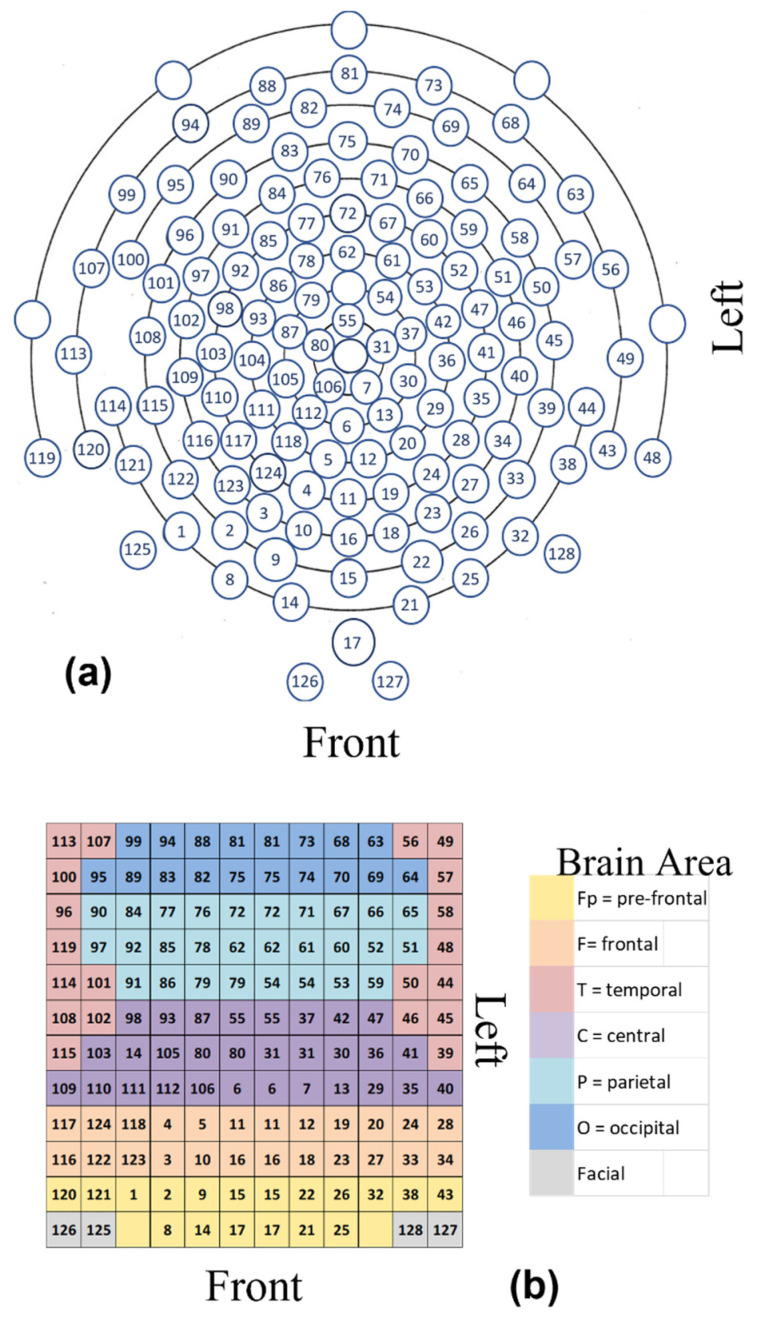
Illustration of electrode arrangements; (**a**) displays the EEG electrodes positions and numbers, and (**b**) shows a representation matrix (A) of 12 × 12 for the EEG 128-channel sensor net over the whole scalp, with some electrode positions repeated to fill the matrix array. This matrix presents the areas of the brain according to the corresponding color, as shown in the legend to the right of the matrix. The left and right hemispheres divide the matrix into 2 equal and symmetrical parts of 2 sub matrices of 12 × 6 each. Positions A_1,3_ and A_1,10_, starting from the bottom left, are empty.

**Figure 2 sensors-23-01293-f002:**
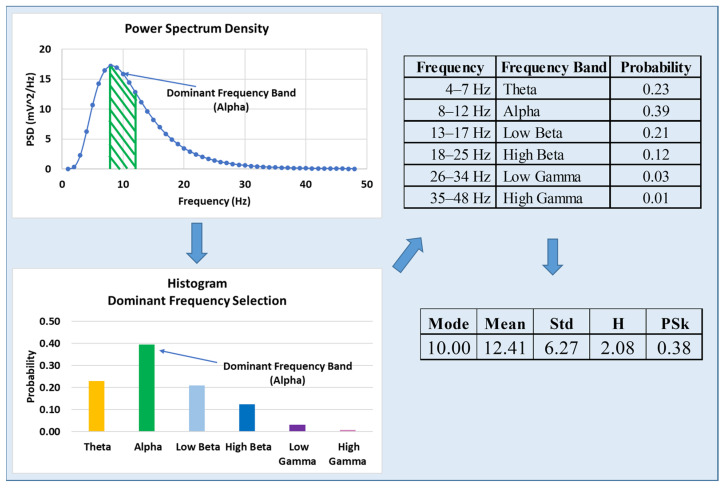
Illustration of the proposed approach to derive the H and PSk values from the power spectrum, and specifically PSk from the dominant frequency band; the frequency resolution in the computation of the power spectrum is 1 Hz. The numerical values are given as illustrations for typical experiments; details are described in the next section.

**Figure 3 sensors-23-01293-f003:**
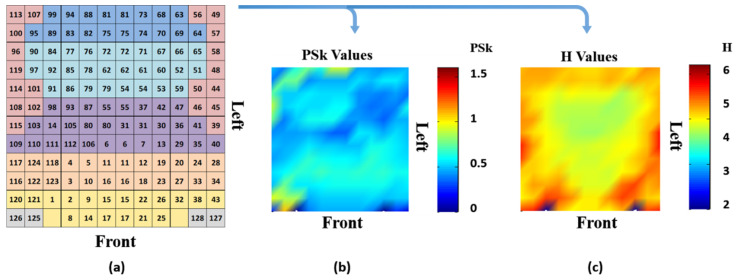
(**a**) Matrix representation of the EEG array; (**b**) a spatial 2D landscape associated with different brain areas, derived from the mean values of PSk, and (**c**) a 2D landscape plot derived from the mean values of H.

**Figure 4 sensors-23-01293-f004:**
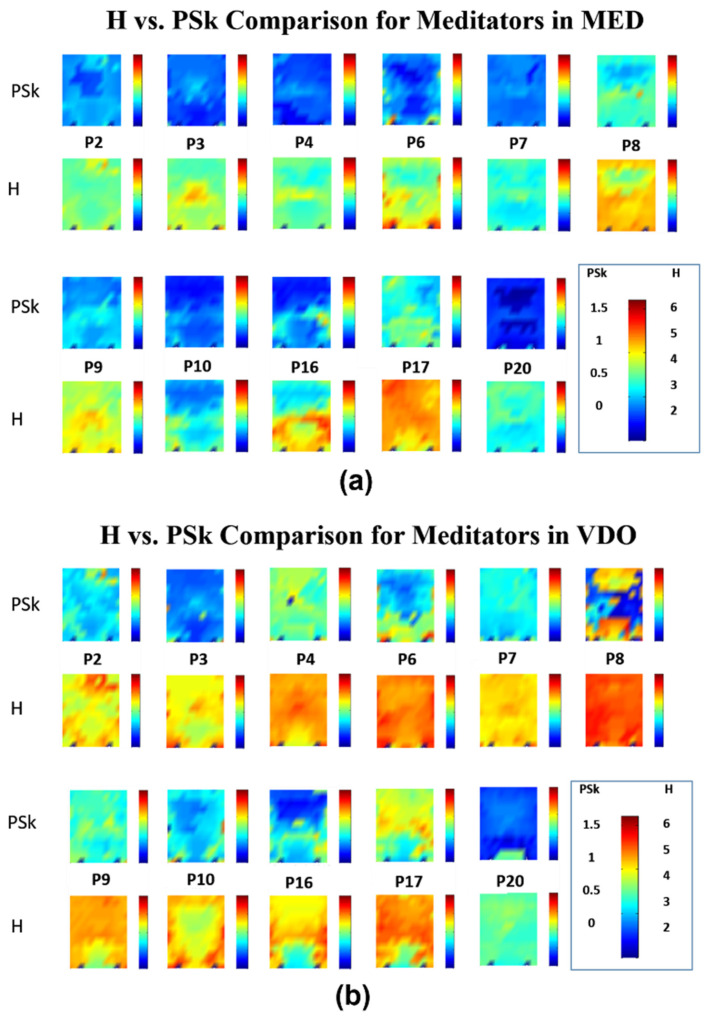
Illustration of the results for eleven (11) *Meditators*; (**a**) values for H (2nd and 4th row) and PSk (1st and 3rd row) in MED modality; (**b**)values for H (2nd and 4th row) and PSk (1st and 3rd row) in VDO modality.

**Figure 5 sensors-23-01293-f005:**
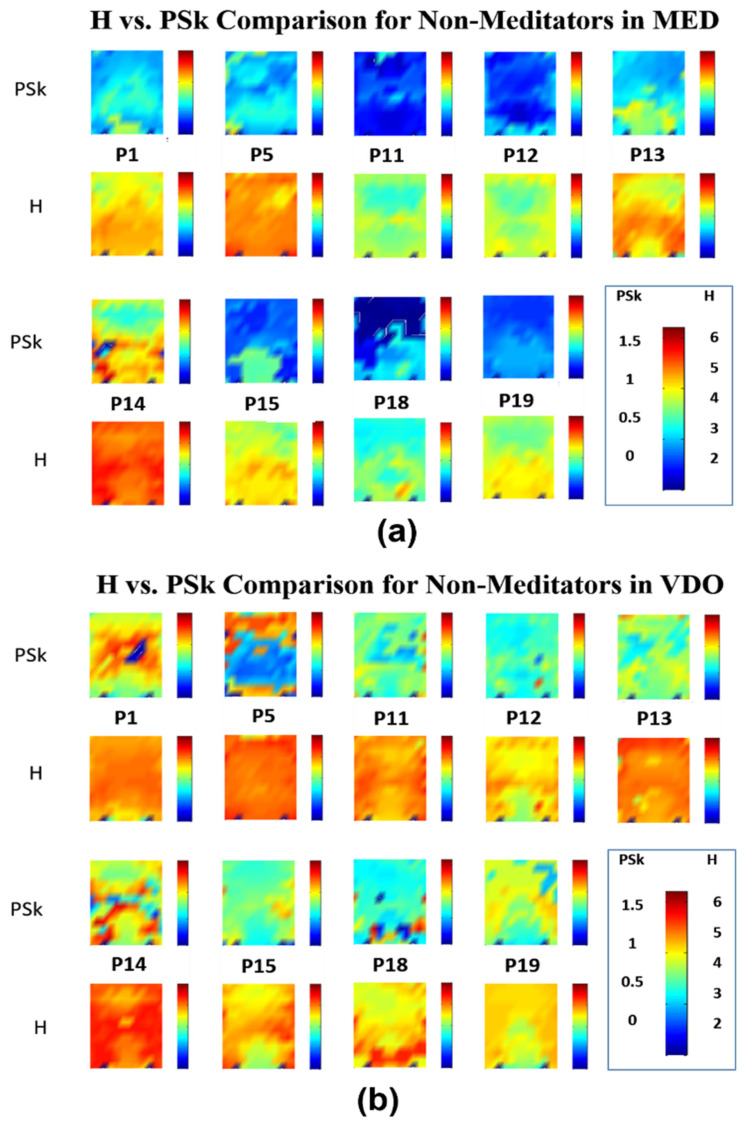
Illustration of the results for nine (9) *Non-Meditators*; (**a**) values for H (2nd and 4th row) and PSk (1st and 3rd row) in MED modality; (**b**) values for H (2nd and 4th row) and PSk (1st and 3rd row) in VDO modality.

**Figure 6 sensors-23-01293-f006:**
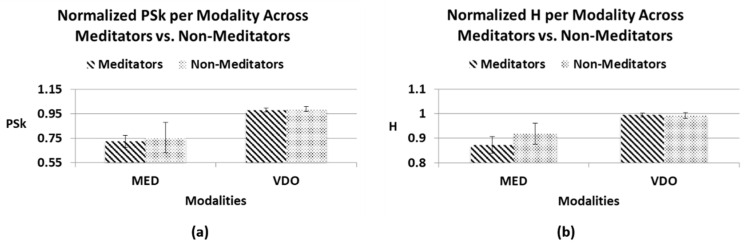
Normalized overall mean values for the statistical indices with confidence intervals for both groups (Meditators, Non-Meditators), in both modalities (MED, VDO); (**a**) index PSk; (**b**) index H.

**Figure 7 sensors-23-01293-f007:**
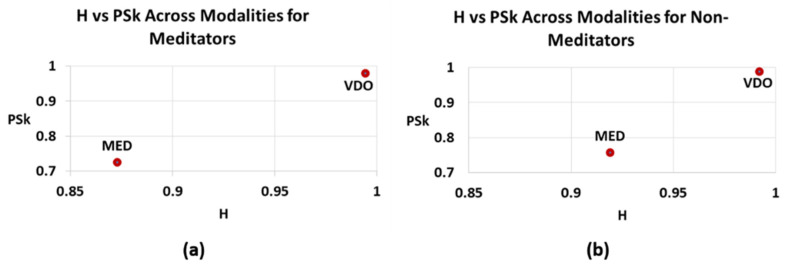
Overall normalized mean values for H (X-axis) vs. PSk (Y-axis) for both modalities (MED, VDO); (**a**) *Meditators*; (**b**) *Non-Meditators*.

**Figure 8 sensors-23-01293-f008:**
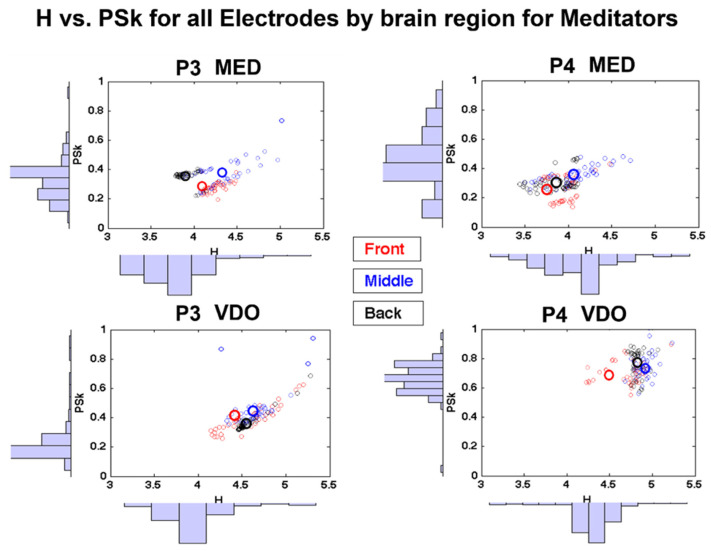
Shows scatter plots of H (x-axis) vs. PSk (y-axis) and their corresponding histograms for the mean values, ${\bar{H}}_{e}^{p,m}and\;{\bar{PSk}}_{e}^{p,m},\;$computed for each of the 128 electrodes (e =1128), for participants P3 (*p* = 3) and P4 (*p* = 4) in MED (m = 1) and VDO (m = 2). Each electrode is represented by a small circle, and its color identifies a brain area, where red corresponds to the frontal area, blue to the central and black to the posterior area. The larger circles represent the overall mean value per brain area.

**Figure 9 sensors-23-01293-f009:**
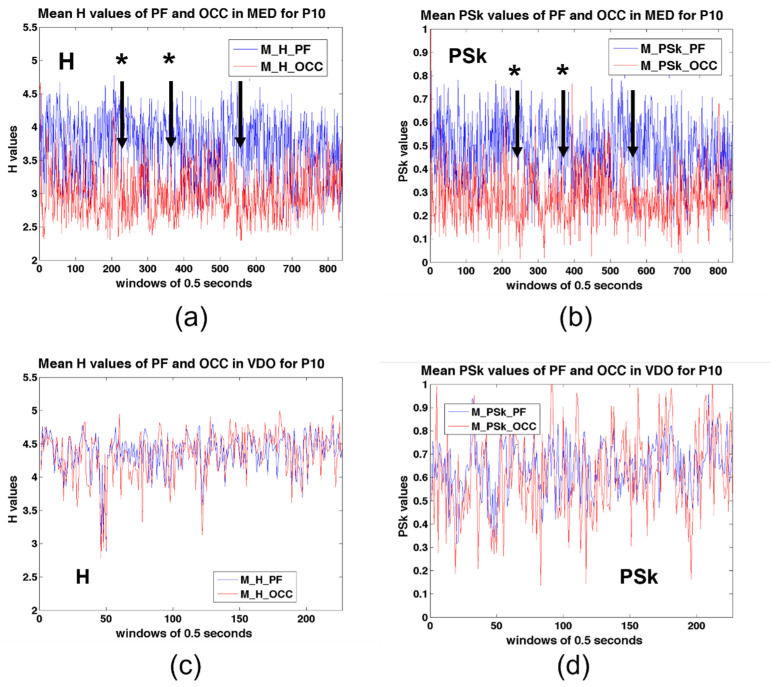
Mean values for the H and PSk time series in time steps of 0.5 s for P10, both for the PF and OCC areas of the brain. (**a**) H for the PF and OCC areas in MED, (**b**) PSk for the PF and OCC areas in MED, (**c**) H for the PF and OCC areas in VDO and (**d**) PSK for the PF and OCC areas in VDO. The vertical arrows with stars on subplots (**a**,**b**) indicate possible anticorrelation events between the occipital and prefrontal areas.

**Figure 10 sensors-23-01293-f010:**
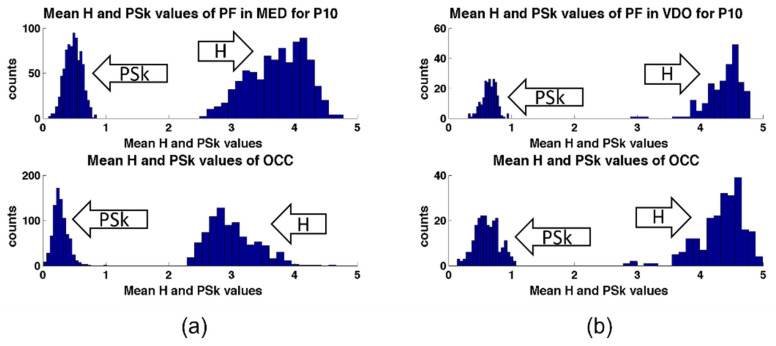
Histograms of H and PSk in various modalities and brain areas of P10; (**a**) (top) histograms for H and PSk for the PF region for P10 in MED, and (bottom) histograms for H and PSk for the OCC region in MED. (**b**) (top) histograms for H and PSk for the PF region for P10 in VDO, and (bottom) histograms for H and PSk for the OCC region in VDO.

**Figure 11 sensors-23-01293-f011:**
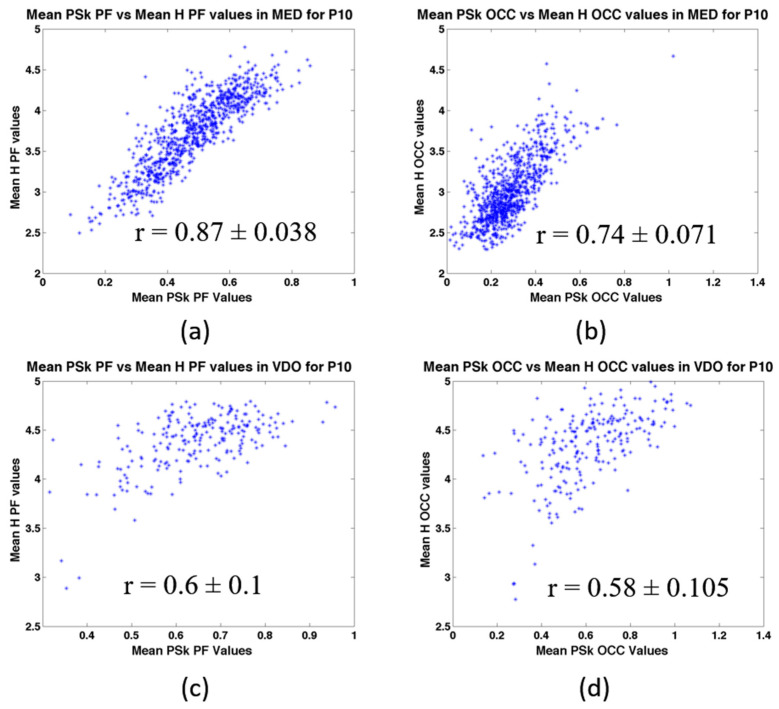
Shows coefficients of correlation between H and PSk for: (**a**) PF area in MED for P10, (**b**) OCC area in MED for P10, (**c**) PF area in VDO for P10 and (**d**), (**b**) OCC area in VDO for P10.

**Figure 12 sensors-23-01293-f012:**
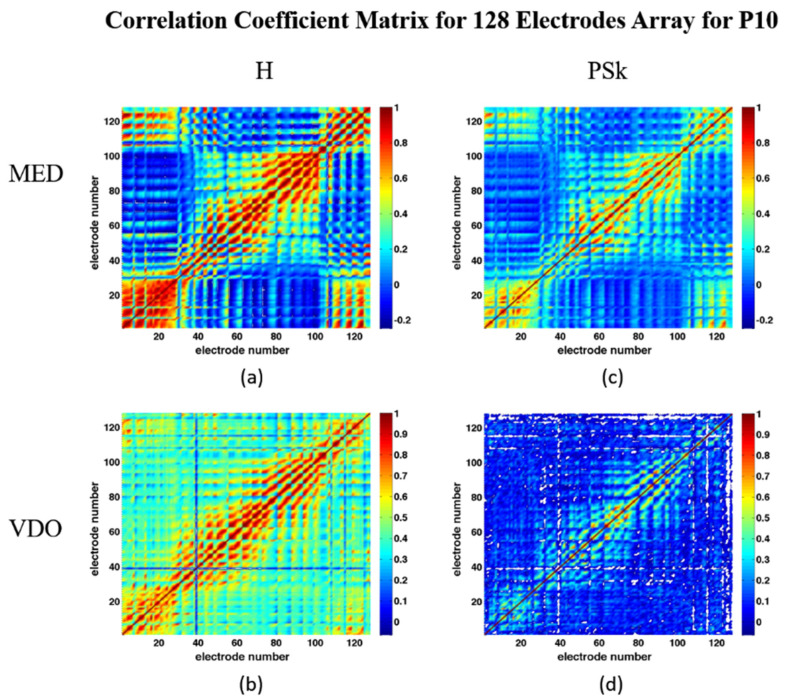
Correlation matrix for: (**a**) H in MED, (**b**) H in VDO, (**c**) PSk in MED, (**d**) PSk in VDO. Please note that, based on Figure 1b, the PF region is described by electrodes with relatively low serial numbers starting with 1 and extending to lower/mid 30′s, and some with very high numbers, while OCC electrodes have numbers starting from around 63 and up to about 99.

**Table 1 sensors-23-01293-t001:** H mean values (${\bar{H}}_{1}{}^{1},\;{\bar{H}}_{1}{}^{2}$, ${\bar{H}}_{2}{}^{1}\;and\;{\bar{H}}_{2}{}^{2}$) and their confidence intervals, for both groups in both modalities.

Group/Modality	MED	VDO
Meditator	0.87 ± 0.033	0.99 ± 0.007
Non-Meditator	0.92 ± 0.043	0.98 ± 0.016

**Table 2 sensors-23-01293-t002:** PSk mean values (${\bar{PSk}}_{1}{}^{1},\;{\bar{PSk}}_{1}{}^{2}$, ${\bar{PSk}}_{2}{}^{1}\;and\;{\bar{PSk}}_{2}{}^{2}$) and their confidence intervals, for both groups in both modalities.

Group/Modality	MED	VDO
Meditator	0.72 ± 0.049	0.98 ± 0.016
Non-Meditator	0.76 ± 0.124	0.99 ± 0.021

**Table 3 sensors-23-01293-t003:** *p*-values of the t-tests with unequal variances, regarding the hypothesis H0: μ1 = μ2; results are given for the MED vs. VDO modalities, respectively, for all brain regions.

Test	*p*-Value	H0: μ1 = μ2
Meditators vs. Non-Meditators for MED (H)	0.033	Reject
Meditators vs. Non-Meditators for MED (PSk)	0.255	Accept
Meditators vs. Non-Meditators for VDO (H)	0.5	Accept
Meditators vs. Non-Meditators for VDO (PSk)	0.955	Accept

**Table 4 sensors-23-01293-t004:** Shows the coefficient of correlation (r) between H and PSk in the different modalities of MED and VDO, for both the PF and OCC areas of the brain, for P10.

Modality and Brain Region	Mean Value Correlation Coefficient (r)	Lower Bound	Upper Bound
MED-PF	0.8649	0.832	0.908
MED-OCC	0.7355	0.669	0.811
VDO-PF	0.6011	0.5	0.7
VDO-OCC	0.5792	0.475	0.685

**Table 5 sensors-23-01293-t005:** Displays ***p***-values and results for various unequal variance *t*-tests of hypothesis, where H0: μ1 = μ2, allowing for a comparison between the MED vs. VDO modalities for the PF and OCC areas based on the H index.

Test	*p*-Value	H0: μ1 = μ2
Meditators vs. Non-Meditators for MED (PF)	0.0218	Reject
Meditators vs. Non-Meditators for MED (OCC)	0.0290	Reject
Meditators vs. Non-Meditators for VDO (PF)	0.0841	Accept
Meditators vs. Non-Meditators for VDO (OCC)	0.1718	Accept

**Table 6 sensors-23-01293-t006:** Displays *p*-values and results for various unequal variance *t*-tests of hypothesis, where H0: μ1 = μ2, allowing for a comparison between the MED vs. VDO modalities for the PF and OCC areas based on the PSk index.

Test	*p*-Value	H0: μ1 = μ2
Meditators vs. Non-Meditators for MED (PF)	0.0437	Reject
Meditators vs. Non-Meditators for MED (OCC)	0.0936	Accept
Meditators vs. Non-Meditators for VDO (PF)	0.0112	Reject
Meditators vs. Non-Meditators for VDO (OCC)	0.8738	Accept

## Data Availability

The EEG datasets generated for this study may be available upon request, provided that the proper approval is granted by Ian J. Kirk, Head of Ian J. Kirk’s Lab at the Centre for Brain Research at the University of Auckland in New Zealand.

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
