# Peer review of "Analysis of Meditation vs. Sensory Engaged Brain States Using Shannon Entropy and Pearson’s First Skewness Coefficient Extracted from EEG Data"

_sensors, 2023, doi:10.3390/s23031293_

Round 1

Reviewer 1 Report

sensors-2102841-peer-review

Research Paper

Analysis of Meditation vs. Sensory Engaged Brain States using

Shannon Entropy and Pearson’s First Skewness Coefficient

Extracted from EEG Data

J. J. Joshua Davis1, Robert Kozma,  Florian Schübeler

Manuscript Summary:

The study is about meditative states and to show different brain dynamics than other more

engaged states. The study was based on the concept  to derive the Shannon Entropy (H) and Pearson’s 1st SkewnessCoefficient (PSk) from the power spectrum for the modalities of Meditation, Relaxation and Video- Watching for eleven (11) Meditators and nine (9) non-Meditators.  H and PSk values were used to analyse and therefore classify different brain states. Then to show significant differences in the brain states based on both qualitative and quantitative analysis.  Shannon Entropy Index (H) and the Pearson’s Skewness Coefficient (PSk) were used to classify brain dynamics for different brain regions.

Congratulations for putting this article together, … well-done. But still, I have few comments.

Well written manuscript.

Well explained concepts.

General comments:

-          The abstract is not well written.  It needs to be made much focused towards the works done.

-          The study was based on some previous work done work the research (5,11, and 49). Therefore, the manuscript is an extension to some previous work. I found some relations and maths equations were extracted from (11) .. but with indication to that.

-          For the EEG experiment, I did not find any information nor explanation to clarify the nature of the experiment and how the (Meditation vs. Sensory Engaged Brain States) were done.  This is an essential part in the experiment to be clarified.

-          There is an issue with clarity of some figures.  In specific figure (1), Figure 1. Illustration of electrode arrangements; (a) displaying the EEG electrodes positions and 132 numbers, … is not clear.  There are other figures that they are not clear as well.  This is an essential issue to be fixed.

-          Some parts in the manuscript are not well referenced, example is at line (473) :

473 intervals at 95% significance [48].

The mathematics for the EEG analysis are well written and well used.

Well written conclusions. 

Well documented references.

Concept of the (309  - 4.2. Detailed Quantitative Analysis of Brain Dynamics), 

This part needs more and further elaboration on the Brain Dynamics. 

Typo error and a need to check:

The manuscript needs a final check for some typo errors, examples …

-          Line (501).  I full stop is needed before the end of the line.

501 dynamics for different brain regions, different participants and different modalities In the.

-          I found that the term (also) has been repeated 20 times in the manuscript.  (Also) can be replaced by a better word like (In addition), or others ..

Reviewer 2 Report

This paper presents the work on analyzing and classifying mediation and watching video state and experienced meditator and novice. The authors' focus is to investigate Shannon entropy and Person's first skewness to see the distribution of the EEG signals in time series. The analysis was done visually and quantitatively.   The strength of this paper is that this is the first study to explore the entropy and skewness analysis between the experienced meditator and the inexperienced meditator. The discussion is reasonable. The authors recruited participants by themself and collected EEG at two conditions with many electrodes.   I have some concerns as follows. First, the criteria of separation of meditator or non-meditator are ambiguous to me. The author explained it as two years regularly, at least five days a week. How do you confirm it, and do you have any reference to guarantee that they are meditators?   Second, the paper is hard to follow because the section is not well organized. The author mentions qualitative analysis in the result section and then mentions about calculation process in the result section regarding the quantitative analysis in the result section. It isn't easy to follow for readers. This should be summarized in the method section. And also, the result and discussion are difficult to follow. Sometimes the result section includes discussion or interpretation, and vice versa.   Third, I think there are different measures for calculating the EEG signals, such as spectral power, asymmetry index, etc. Why did you select the entropy and skewness in particular? There is no background evidence for selecting these two measures.   Forth, I think one of the goals of the quantitative analysis is to classify the conditions. However, the author reported only the mean and confidence interval. I think the authors also should report classification evaluation, such as statistical testing or classification accuracy.   Fifth, the task used is meditation and watching the video. I understand the first one, but I can not imagine the video. Do you provide an image of the video and the experimental procedure? It could be needed for experiment reproduction. And I think these two conditions are very different regarding stimuli and eye openness. For instance, what do you think about including a condition of open eyes?   Lastly, though this is a minor point, it is difficult to see the number with brackets: like two (2). I don't know if this is written in the guideline of the Sensors, but it is better to be written without brackets, at least.

Reviewer 3 Report

The submitted paper aimed at clarifying differences in brain dynamics in the state of meditation, relaxation and video-watching using the Shannon Entropy (H) and Pearson’s 1st Skewness Coefficient (PSk) estimated from the power spectrum of EEG measured with 128-channel electroencephalography.

My comments on the paper are as follows.

1. Qualitative and quantitative differences between mediators and non-mediators in MED (such as Figure 7) may reflect differences in brain dynamics, which is interesting from a psychophysiological point of view. However, as mentioned by the author between lines (ls.) 288 to 290 in page 10, such differences in H and PSk between MED and VDO may be intensively due to the eyes-open/closure, which do not depend on tasks performed in the present experiment. This point should be carefully discussed and reflected in the revised manuscript.

2. There are concerns about the order effects for MED and VDO tasks. The order of tasks should be counter-balanced in order to avoid a bias of task order which may affect the experimental results.

3. Discussion was mainly based on the analyzed results for Participant 10. However, for corroborating the author’s conjecture, discussion on other participants is also needed and classification of brain dynamics should be performed in terms of the histograms of H and PSk and correlation coefficients, if possible.

Other minor points are also listed:

1. (2. Materials) Time-lengths of MED and VDO each trial should be described.

2. (2. Materials) How was the age structure of the participants in each group?

3. (3.1 Preprocessing) How long was the maximum time difference between two channels in each data sampling?

4. (Figure 1 (a)) Since it is impossible to read the figure of number of channels, the quality of the image should be improved. In addition, the electrode positions look upside down and mirror-based to Figure 1(b).

5. (Figure 2) To which band do the boundary frequency components such as 7.5 Hz and 12.5 Hz components belong?

6. (Figure 9 and ls 406-408, p. 14) Differences in PSk mean values between the frontal and posterior regions seems to be small for all the plots (P3 and P4 and MED and VDO).

7. (Figures 8 and 9) Figures 8 and 9 could be merged to illustrate both the scatter plots with histograms and the regional difference with the mean values.

8. (ls. 454-457, p. 15) Time periods in which both H and PSk are inversely correlated should be shown.

Round 2

Reviewer 2 Report

I find the authors carefully revised the paper according to my comments.

Author Response

Thank you very much for your valuable comments and supporting our work.

Reviewer 3 Report

The revised manuscript has fixed almost all my questions and comments. However, I still have a few comments for the revised manuscript.

1. (In my previous comment for the time difference)

[3. (3.1 Preprocessing) How long was the maximum time difference between two channels in each data sampling?

(Authors’ response) As a general approach, all adjustments and preprocessing apply equally to all channels to guaranty that all signals remain the same length. (line 195-196, p.5)]

The meaning of my question is as follows.

The 128 electrodes dense-array electroencephalography used for EEG recording may have an analog multiplexer to select one analog signal from 128 analog inputs and transmit the selected signal to a sample and hold circuit for analog-to-digital conversion. If so, the time (or phase) difference or certain delay time occurs between digital data each sampling. For example, if there occurs a time difference of t0 between consecutively discretized adjacent two digital data, the maximum time difference in digital data between two channels becomes 127t0 each sampling period. Such a time delay has little impact on the results in the current study, as being included in minor comments, but the basic information in signal acquisition.

2. (line 420, p. 13 and line 670, p. 20) The value 0.01 should be corrected to 0.05.
